# Non-asymptotic approximations of Gaussian networks via second-order Poincaré inequalities

## Abstract

There is a recent and growing interest on large-width asymptotic properties of Gaussian neural networks (NNs), namely NNs whose weights are initialized according to Gaussian distributions. A well-established result is that, as the width goes to infinity, a Gaussian NN converges in distribution to a Gaussian stochastic process, which provides an asymptotic or qualitative Gaussian approximation of the NN. In this paper, we introduce some non-asymptotic or quantitative Gaussian approximations of Gaussian NNs, quantifying the approximation error with respect to some popular distances for (probability) distributions, e.g. the 1-Wasserstein distance, the total variation distance and the Kolmogorov-Smirnov distance. Our results rely on the use of second-order Gaussian Poincaré inequalities, which provide tight estimates of the approximation error, with optimal rates. This is a novel application of second-order Gaussian Poincaré inequalities, which are well-known in the probabilistic literature for being a powerful tool to obtain Gaussian approximations of general functionals of Gaussian stochastic processes. A generalization of our results to deep Gaussian NNs is discussed.

## 1 Introduction

There is a growing interest on large-width asymptotic properties of Gaussian neural networks (NNs), namely NNs whose weights or parameters are initialized according to Gaussian distributions (Neal, 1996; Williams, 1997; Der & Lee, 2005; Garriga-Alonso et al., 2018; Lee et al., 2018; Matthews et al., 2018; Novak et al., 2018; Antognini, 2019; Hanin, 2019; Yang, 2019; Aitken & Gur-Ari, 2020; Andreassen & Dyer, 2020; Bracale et al., 2021; Eldan et al., 2021; Basteri & Trevisan, 2022). Let $\mathcal{N}(\mu, \sigma^2)$ be a Gaussian distribution with mean $\mu$ and variance $\sigma^2$, and consider: i) an input $\boldsymbol{x} \in \mathbb{R}^d$, with $d \geq 1$; ii) a collection of (random) weights $\theta = \{w_i^{(0)}, w, b_i^{(0)}, b\}_{i \geq 1}$ such that $w_{i,j}^{(0)} \stackrel{d}{=} w_j$, with the $w_{i,j}^{(0)}$'s being independent and identically distributed as $\mathcal{N}(0, \sigma_w^2)$, and $b_i^{(0)} \stackrel{d}{=} b$, with the $b_i^{(0)}$'s being independent and identically distributed as $\mathcal{N}(0, \sigma_b^2)$ for $\sigma_w^2, \sigma_b^2 > 0$; iii) an activation function $\tau : \mathbb{R} \to \mathbb{R}$. Then, a (fully connected feed-forward) Gaussian NN is defined as follows:

$$f_{\boldsymbol{x}}(n)[\tau, n^{-1/2}] = b + \frac{1}{n^{1/2}} \sum_{j=1}^{n} w_j \tau(\langle w_j^{(0)}, \boldsymbol{x} \rangle_{\mathbb{R}^d} + b_j^{(0)}), \tag{1}$$

with $n^{-1/2}$ being a scaling factor. Neal (1996) characterized the infinitely wide limit of the NN equation 1, showing that, as $n \to +\infty$, for any $\boldsymbol{x} \in \mathbb{R}^d$ the NN $f_{\boldsymbol{x}}(n)[\tau, n^{-1/2}]$ converges in distribution to a Gaussian random variable (RV). That is, as a function of $\boldsymbol{x}$, the infinitely wide limit of the NN is a Gaussian stochastic process. The proof is an application of the classical Central Limit Theorem (CLT), thus relying on minimal assumptions on $\tau$ to ensure that $\mathbb{E}[(g_j(\boldsymbol{x}))^2]$ is finite, where $g_j(\boldsymbol{x}) = w_j \tau(\langle w_j^{(0)}, \boldsymbol{x} \rangle_{\mathbb{R}^d} + b_j^{(0)})$. The result of Neal (1996) has been extended to more general matrix input, i.e. $p > 1$ inputs of dimension $d$, and to deep Gaussian NNs, assuming a "sequential growth" (Der & Lee, 2005) and a "joint growth" (Matthews et al., 2018) of the width over the NN's layers. These results provide asymptotic or qualitative Gaussian approximations of Gaussian NNs, as they do not provide the rate at which the NN converges to the infinitely wide limit.

## 1.1 Our contribution

In this paper, we consider non-asymptotic or quantitative Gaussian approximations of the NN equation 1, quantifying the approximation error with respect to some popular distances for (probability) distributions. To introduce our results, let $d_{W_1}$ be the 1-Wasserstein distance and consider a Gaussian NN with a 1-dimensional unitary input, i.e. $d = 1$ and $x = 1$, unit variance's weight, i.e. $\sigma_w^2 = 1$, and no biases, i.e. $b_i^{(0)} = b = 0$ for any $i \geq 1$. Under this setting, our result states as follows: if $\tau \in C^2(\mathbb{R})$ such $\tau$ and its first and second derivatives are bounded above by the linear envelope $a + b|x|^\gamma$, for $a, b, \gamma > 0$, and $N \sim \mathcal{N}(0, \sigma^2)$ with $\sigma^2$ being the variance of the NN, then for any $n \geq 1$

$$d_{W_1}(f_1(n)[\tau, n^{-1/2}], N) \leq \frac{K_{\sigma^2}}{n^{1/2}}, \tag{2}$$

with $K_{\sigma^2}$ being a constant that can be computed explicitly. The polynomial envelope assumption is not new in the study of large-width properties of Gaussian NNs (Matthews et al., 2018; Yang, 2019), and it is critical to achieve the optimal rate $n^{-1/2}$ in the estimate equation 2 of the approximation error. In general, we show that an approximation analogous to equation 2 holds true for the Gaussian NN equation 1, with the approximation being with respect to the 1-Wasserstein distance, the total variation distance and the Kolmogorov-Smirnov distance. Our results rely on the use of second-order Gaussian Poincaré inequalities, or simply second-order Poincaré inequalities, first introduced in Chatterjee (2009) and Nourdin et al. (2009) as a powerful tool to obtain Gaussian approximation of general functionals of Gaussian stochastic processes. Here, we make use of some refinements of second-order Poincaré inequalities developed in Vidotto (2020), which have the advantage of providing tight estimates of the approximation error, with (presumably) optimal rates. An extension of equation 2 is presented for Gaussian NNs with $p > 1$ inputs, whereas a generalization of our results to deep Gaussian NNs is discussed with respect to the "sequential growth" and the "joint growth" of the width over the NN's layers.

## 1.2 Related work

While there exists a vast literature on infinitely wide limits of Gaussian NNs, as well as their corresponding asymptotic approximations, only a few recent works have investigated non-asymptotic approximations of Gaussian NNs. To the best of our knowledge, the work of Eldan et al. (2021) is the first to consider the problem of non-asymptotic approximations of Gaussian NNs, focusing on NNs with Gaussian distributed weights $w_{i,j}$'s and Rademacher distributed weights $w_i$'s. For such a class of NNs, they established a quantitative CLT in an infinite-dimensional functional space, metrized with the Wasserstein distance, providing rates of convergence to a Gaussian stochastic process. For deep Gaussian NNs (Der & Lee, 2005; Matthews et al., 2018), the work of Basteri & Trevisan (2022) first established a quantitative CLT in the 2-Wasserstein distance, providing the rate at which a deep Gaussian NN converges to its infinitely wide limit. Such a result relies on basic properties of the Wasserstein distance, which allow for a quantitatively tracking the hidden layers and yield a proof by induction, with the triangular inequality being applied to obtain independence from the previous layers. See Favaro et al. (2022) for an analogous result in the sup-norm distance. Our work is close to that of Basteri & Trevisan (2022), in the sense that we deal with NNs for which all the weights are initialized according to Gaussian distributions, and we consider their approximations through Gaussian RVs. The novelty of our work lies on the use of second-order Poincaré inequalities, which allow reducing the problem to a direct computation of the gradient and Hessian of the NN, and provide estimates of the approximation error with optimal rate, and tight constants, with respect to other distances than sole Wasserstein distance. This is the first to make use of second-order Poincaré inequalities as a tool to obtain non-asymptotic Gaussian approximations of Gaussian NNs.

## 1.3 Organization of the paper

The paper is structured as follows. In Section 2 we present an overview on second-order Poincaré inequalities, recalling some of the main results of Vidotto (2020) that are critical to prove our non-asymptotic Gaussian approximations of Gaussian NNs. Section 3 contains the non-asymptotic Gaussian approximation of the NN equation 1, as well as its extension for the NN equation 1 with $p > 1$ inputs, where Section 4 contains some numerical illustrations of our approximations. In Section 5 we discuss the extension of our results to deep Gaussian NNs, and we present some directions for future research.

# 2 Preliminaries on second-order Poincaré inequalities

Let $(\Omega, \mathcal{F}, \mathbb{P})$ be a generic probability space on which all the RVs are assumed to be defined. We denote by $\perp\!\!\!\perp$ the independence between RVs, and we make use of the acronym "iid" to refer to RVs that are independent and identically distributed and by $\|X\|_{L^q} := (\mathbb{E}[X^q])^{1/q}$ the $L^q$ norm of the RV $X$. In this work, we consider some popular distances between (probability) distributions of real-valued RVs. Let $X$ and $Y$ be two RVs in $\mathbb{R}^d$, for some $d \geq 1$. We denote by $d_{W_1}$ the 1-Wasserstein distance, that is,

$$d_{W_1}(X, Y) = \sup_{h \in \mathscr{H}} |\mathbb{E}[h(X)] - \mathbb{E}[h(Y)]|,$$

where $\mathscr{H}$ is the class of all functions $h : \mathbb{R}^d \to \mathbb{R}$ such that it holds true that $\|h\|_{\text{Lip}} \leq 1$, with $\|h\|_{\text{Lip}} = \sup_{x,y \in \mathbb{R}^d, x \neq y} |h(x) - h(y)|/\|x - y\|_{\mathbb{R}^d}$. We denote by $d_{TV}$ the total variation distance, that is,

$$d_{TV}(X, Y) = \sup_{B \in \mathscr{B}(\mathbb{R}^m)} |\mathbb{P}(X \in B) - \mathbb{P}(Y \in B)|,$$

where $\mathscr{B}\left(\mathbb{R}^d\right)$ is the Borel $\sigma$-field of $\mathbb{R}^d$. Finally, we denote by $d_{KS}$ the Kolmogorov-Smirnov distance, i.e.

$$d_{KS}(X, Y) = \sup_{z_1, \ldots, z_d \in \mathbb{R}} |\mathbb{P}\left(X \in \times_{i=1}^d (-\infty, z_i]\right) - \mathbb{P}\left(Y \in \times_{i=1}^d (-\infty, z_i]\right)|.$$

We recall the following interplays between some of the above distances: i) $d_{KS}(\cdot, \cdot) \leq d_{TV}(\cdot, \cdot)$; ii) if $X$ is a real-valued RV and $N \sim \mathcal{N}(0, 1)$ is the standard Gaussian RV then $d_{KS}(X, N) \leq 2\sqrt{d_{W_1}(X, N)}$.

Second-order Poincaré inequalities provide a useful tool for Gaussian approximation of general functionals of Gaussian fields (Chatterjee, 2009; Nourdin et al., 2009). See also Nourdin & Peccati (2012) and references therein for a detailed account. For our work, it is useful to recall some results developed in Vidotto (2020), which provide improved versions of the second-order Poincaré inequality first introduced in Chatterjee (2009) for random variables and then extended in Nourdin et al. (2009) to general infinite-dimensional Gaussian fields. Let $N \sim \mathcal{N}(0, 1)$. Second-order Poincaré inequalities can be seen as an iteration of the so-called Gaussian Poincaré inequality, which states that

$$\text{Var}[f(N)] \leq \mathbb{E}[f'(N)^2] \tag{3}$$

for every differentiable function $f : \mathbb{R} \to \mathbb{R}$, a result that was first discovered in a work by Nash (1956) and then reproved by Chernoff (1981). The inequality equation 3 implies that if the $L^2$ norm of the RV $f'(N)$ is small, then so are the fluctuations of the RV $f(N)$. The first version of a second-order Poincaré inequality was obtained in Chatterjee (2009), where it is proved that one can iterate equation 3 in order to assess the total variation distance between the distribution of $f(N)$ and the distribution of a Gaussian RV with matching mean and variance. The precise result is stated in the following theorem.

**Theorem 2.1** (Chatterjee (2009) - second-order Poincaré inequality). *Let $X \sim \mathcal{N}(0, I_{d \times d})$. Take any $f \in C^2(\mathbb{R}^d)$, and $\nabla f$ and $\nabla^2 f$ denote the gradient of $f$ and Hessian of $f$, respectively. Suppose that $f(X)$ has a finite fourth moment, and let $\mu = \mathbb{E}[f(X)]$ and $\sigma^2 = \text{Var}[f(X)]$. Let $N \sim \mathcal{N}(\mu, \sigma^2)$ then*

$$d_{TV}(f(X), N) \leq \frac{2\sqrt{5}}{\sigma^2} \left\{\mathbb{E}\left[\|\nabla f(X)\|_{\mathbb{R}^d}^4\right]\right\}^{1/4} \left\{\mathbb{E}\left[\|\nabla^2 f(X)\|_{op}^4\right]\right\}^{1/4}, \tag{4}$$

*where $\|\cdot\|_{op}$ stands for the operator norm of the Hessian $\nabla^2 f(X)$ regarded as a random $d \times d$ matrix.*

Nourdin et al. (2009) pointed out that the Stein-type inequalities that lead to equation 4 are special instances of a more general class of inequalities, which can be obtained by combining Stein's method and Malliavin calculus on an infinite-dimensional Gaussian space. In particular, Nourdin et al. (2009) obtained a general version of equation 4, involving functionals of arbitrary infinite-dimensional Gaussian fields. Both equation 4 and its generalization in Nourdin et al. (2009) are known to give suboptimal rates of convergence. This is because, in general, it is not possible to obtain an explicit computation of the expectation of the operator norm involved in the estimate of total variation distance, which leads to move further away from the distance in distribution and use bounds on the operator norm instead of computing it directly. To overcome this drawback, Vidotto (2020) adapted to the Gaussian setting an approach recently developed in Last et al. (2016) to obtain second-order Poincaré inequalities for Gaussian approximation of

Poisson functionals, yielding estimates of the approximation error that are (presumably) optimal. The next theorem states Vidotto (2020, Theorem 2.1) for the special case of a function $f(X)$, with $f \in C^2(\mathbb{R}^d)$ such that its partial derivatives have sub-exponential growth, and $X \sim \mathcal{N}(0, I_{d \times d})$. See Appendix A for an overview of Vidotto (2020, Theorem 2.1).

**Theorem 2.2** (Vidotto (2020) - 1-dimensional second-order Poincaré inequality). *Let $F = f(X)$, for some $f \in C^2(\mathbb{R}^d)$, and $X \sim \mathcal{N}(0, I_{d \times d})$ such that $E[F] = 0$ and $E[F^2] = \sigma^2$. Let $N \sim \mathcal{N}(0, \sigma^2)$, then*

$$d_M(F, N) \le c_M \sqrt{\sum_{l,m=1}^{d} \left\{ \mathbb{E}\left[ \left( \langle \nabla_{l,\cdot}^2 F, \nabla_{m,\cdot}^2 F \rangle \right)^2 \right] \right\}^{1/2} \left\{ \mathbb{E}\left[ (\nabla_l F \nabla_m F)^2 \right] \right\}^{1/2}}, \tag{5}$$

*where $\langle \cdot, \cdot \rangle$ is the scalar product, $M \in \{TV, KS, W_1\}$, $c_{TV} = \frac{4}{\sigma^2}, c_{KS} = \frac{2}{\sigma^2}, c_{W_1} = \sqrt{\frac{8}{\sigma^2 \pi}}$ and $\nabla_{i,\cdot}^2 F$ is the $i$-th row of the Hessian matrix of $F = f(X)$ while $\nabla_i F$ is the $i$-th element of the gradient of $F$.*

The next theorem generalizes Theorem 2.2 to multidimensional functionals. In particular, for any $p > 1$, the next theorem states Vidotto (2020, Theorem 2.3) for the special case of a function $(f_1(X), \dots, f_p(X))$, with $f_1, \dots, f_p \in C^2(\mathbb{R}^d)$ such that its partial derivatives have sub-exponential growth, and $X \sim \mathcal{N}(0, I_{d \times d})$. See Appendix A for a brief overview of Vidotto (2020, Theorem 2.3).

**Theorem 2.3** (Vidotto (2020) - $p$-dimensional second-order Poincaré inequality). *For any $p > 1$ let $(F_1, \dots, F_p) = (f_1(X), \dots, f_p(X))$, for some $f_1, \dots, f_p \in C^2(\mathbb{R}^d)$, and $X \sim \mathcal{N}(0, I_{d \times d})$ such that $E[F_i] = 0$ for $i = 1, \dots, p$ and $E[F_i F_j] = c_{ij}$ for $i, j = 1, \dots, p$, with $C = \{c_{ij}\}_{i,j=1,\dots,p}$ being a symmetric and positive definite matrix, i.e. a variance-covariance matrix. Let $N \sim \mathcal{N}(0, C)$, then*

$$d_{W_1}(F, N) \tag{6}$$
$$\le 2\sqrt{p} \left\| C^{-1} \right\|_2 \|C\|_2 \sqrt{\sum_{i,k=1}^{p} \sum_{l,m=1}^{d} \left\{ \mathbb{E}\left[ \left( \langle \nabla_{l,\cdot}^2 F_i, \nabla_{m,\cdot}^2 F_i \rangle \right)^2 \right] \right\}^{1/2} \left\{ \mathbb{E}\left[ (\nabla_l F_k \nabla_m F_k)^2 \right] \right\}^{1/2}}$$

*where $\|\cdot\|_2$ is the spectral norm of a matrix.*

## 3 Main results

In this section, we present the main result of the paper, namely a non-asymptotic Gaussian approximation of the NN equation 1, quantifying the approximation error with respect to the 1-Wasserstein distance, the total variation distance and the Kolmogorov-Smirnov distance. It is useful to start with the simple setting of a Gaussian NN with a 1-dimensional unitary input, i.e. $d = 1$ and $x = 1$, unit variance's weight, i.e. $\sigma_w^2 = 1$, and no biases, i.e. $b_i^{(0)} = b = 0$ for any $i \ge 1$. That is, we consider the NN

$$F := f_1(n)[\tau, n^{-1/2}] = \frac{1}{n^{1/2}} \sum_{j=1}^{n} w_j \tau(w_j^{(0)}). \tag{7}$$

By means of a straightforward calculation, one has $\mathbb{E}[F] = 0$ and $\text{Var}[F] = \mathbb{E}_{Z \sim \mathcal{N}(0,1)}[\tau^2(Z)]$. As $F$ in equation 7 is a function of independent standard Gaussian RVs, Theorem 2.2 can be applied to approximate $F$ with a Gaussian RV with the same mean and variance as $F$, quantifying the approximation error.

**Theorem 3.1.** *Let $F$ be the NN equation 7 with $\tau \in C^2(\mathbb{R})$ such that $|\tau(x)| \le a + b|x|^\gamma$ and $\left| \frac{d^l}{dx^l} \tau(x) \right| \le a + b|x|^\gamma$ for $l = 1, 2$ and some $a, b, \gamma \ge 0$. If $N \sim \mathcal{N}(0, \sigma^2)$ with $\sigma^2 = \mathbb{E}_{Z \sim \mathcal{N}(0,1)}[\tau^2(Z)]$, then for any $n \ge 1$*

$$d_M(F, N) \le \frac{c_M}{\sqrt{n}} \sqrt{3(1 + \sqrt{2})} \cdot \left\| a + b|Z|^\gamma \right\|_{L_4}^2, \tag{8}$$

*where $Z \sim \mathcal{N}(0, 1)$, $M \in \{TV, KS, W_1\}$, with corresponding constants $c_{TV} = 4/\sigma^2$, $c_{KS} = 2/\sigma^2$, and $c_{W_1} = \sqrt{8/\sigma^2 \pi}$.*

*Proof.* To apply Theorem 2.2, we start by computing some first and second order partial derivatives. That is,

$$
\begin{cases}
\frac{\partial F}{\partial w_j} = n^{-1/2}\tau(w_j^{(0)}) \\[2mm]
\frac{\partial F}{\partial w_j^{(0)}} = n^{-1/2}w_j\tau'(w_j^{(0)}) \\[2mm]
\nabla^2_{w_j,w_i}F = 0 \\[2mm]
\nabla^2_{w_j,w_i^{(0)}}F = n^{-1/2}\tau'(w_j^{(0)})\delta_{ij} \\[2mm]
\nabla^2_{w_j^{(0)},w_i^{(0)}}F = n^{-1/2}w_j\tau''(w_j^{(0)})\delta_{ij}
\end{cases}
$$

with $i,j = 1 \dots n$. Then, by a direct application of Theorem 2.2, we obtain the following preliminary estimate

$$
d_M(F,N) \leq c_M \Bigg\{ \sum_{j=1}^n 2 \left\{ \mathbb{E}\left[\left(\langle \nabla^2_{w_j,\cdot}F, \nabla^2_{w_j^{(0)},\cdot}F\rangle\right)^2\right] \mathbb{E}\left[\left(\frac{\partial F}{\partial w_j}\frac{\partial F}{\partial w_j^{(0)}}\right)^2\right]\right\}^{1/2}
$$
$$
+ \left\{ \mathbb{E}\left[\left(\langle \nabla^2_{w_j,\cdot}F, \nabla^2_{w_j,\cdot}F\rangle\right)^2\right] \mathbb{E}\left[\left(\frac{\partial F}{\partial w_j}\frac{\partial F}{\partial w_j}\right)^2\right]\right\}^{1/2}
$$
$$
+ \left\{ \mathbb{E}\left[\left(\langle \nabla^2_{w_j^{(0)},\cdot}F, \nabla^2_{w_j^{(0)},\cdot}F\rangle\right)^2\right] \mathbb{E}\left[\left(\frac{\partial F}{\partial w_j^{(0)}}\frac{\partial F}{\partial w_j^{(0)}}\right)^2\right]\right\}^{1/2} \Bigg\}^{1/2},
$$

which can be further developed. In particular, we can write the right-hand side of the previous estimate as

$$
c_M \Bigg\{ \sum_{j=1}^n 2 \left\{ \mathbb{E}\left[\left(\frac{1}{n}w_j\tau'\left(w_j^{(0)}\right)\tau''\left(w_j^{(0)}\right)\right)^2\right] \mathbb{E}\left[\left(\frac{1}{n}w_j\tau\left(w_j^{(0)}\right)\tau'\left(w_j^{(0)}\right)\right)^2\right]\right\}^{1/2}
$$
$$
+ \left\{ \mathbb{E}\left[\left(\frac{1}{\sqrt{n}}\tau'\left(w_j^{(0)}\right)\right)^4\right] \mathbb{E}\left[\left(\frac{1}{\sqrt{n}}\tau\left(w_j^{(0)}\right)\right)^4\right]\right\}^{1/2}
$$
$$
+ \left\{ \mathbb{E}\left[\left(\frac{1}{n}\left\{\tau'\left(w_j^{(0)}\right)\right\}^2 + \frac{1}{n}w_j^2\left\{\tau''\left(w_j^{(0)}\right)\right\}^2\right)^2\right] \mathbb{E}\left[\left(\frac{1}{\sqrt{n}}w_j\tau'\left(w_j^{(0)}\right)\right)^4\right]\right\}^{1/2} \Bigg\}^{1/2}
$$
$$
\overset{(\mathbb{E}[w_j^2]=1)}{=} \frac{c_M}{n} \Bigg\{ \sum_{j=1}^n 2 \left\{ \mathbb{E}\left[\left(\tau'\left(w_j^{(0)}\right)\tau''\left(w_j^{(0)}\right)\right)^2\right] \mathbb{E}\left[\left(\tau\left(w_j^{(0)}\right)\tau'\left(w_j^{(0)}\right)\right)^2\right]\right\}^{1/2}
$$
$$
+ \left\{ \mathbb{E}\left[\left(\tau'\left(w_j^{(0)}\right)\right)^4\right] \mathbb{E}\left[\left(\tau\left(w_j^{(0)}\right)\right)^4\right]\right\}^{1/2}
$$
$$
+ \left\{ \mathbb{E}\left[\left(\left\{\tau'\left(w_j^{(0)}\right)\right\}^2 + w_j^2\left\{\tau''\left(w_j^{(0)}\right)\right\}^2\right)^2\right] \mathbb{E}\left[\left(w_j\tau'\left(w_j^{(0)}\right)\right)^4\right]\right\}^{1/2} \Bigg\}^{1/2}
$$
$$
\overset{(iid)}{=} \frac{c_M}{\sqrt{n}} \Bigg\{ 2 \left\{ \mathbb{E}\left[\left(\tau'(Z)\tau''(Z)\right)^2\right] \mathbb{E}\left[\left(\tau(Z)\tau'(Z)\right)^2\right]\right\}^{1/2}
$$
$$
+ \left\{ \mathbb{E}\left[\left(\tau'(Z)\right)^4\right] \mathbb{E}\left[\left(\tau(Z)\right)^4\right]\right\}^{1/2}
$$
$$
+ \left\{ \mathbb{E}\left[\left(\left\{\tau'(Z)\right\}^2 + w_j^2\left\{\tau''(Z)\right\}^2\right)^2\right] \mathbb{E}\left[\left(w_j\tau'(Z)\right)^4\right]\right\}^{1/2} \Bigg\}^{1/2}
$$
$$
\overset{(iid)}{=} \frac{c_M}{\sqrt{n}} \Bigg\{ 2 \left\{ \mathbb{E}\left[\left(\tau'(Z)\tau''(Z)\right)^2\right] \mathbb{E}\left[\left(\tau(Z)\tau'(Z)\right)^2\right]\right\}^{1/2}
$$

$$+ \left\{ \mathbb{E}\left[(\tau'(Z))^4\right] \mathbb{E}\left[(\tau(Z))^4\right] \right\}^{1/2}$$

$$+ \left\{ \mathbb{E}\left[\left(\{\tau'(Z)\}^2 + w_j^2\{\tau''(Z)\}^2\right)^2\right] \mathbb{E}\left[(w_j\tau'(Z))^4\right] \right\}^{1/2} \right\}^{1/2}$$

$$= \frac{c_M}{\sqrt{n}} \left\{ 2 \left\{ \mathbb{E}\left[(\tau'(Z)\,\tau''(Z))^2\right] \mathbb{E}\left[(\tau(Z)\,\tau'(Z))^2\right] \right\}^{1/2} \right.$$

$$+ \left\{ \mathbb{E}\left[(\tau'(Z))^4\right] \mathbb{E}\left[(\tau(Z))^4\right] \right\}^{1/2}$$

$$+ \left\{ \left( \mathbb{E}\left[\{\tau'(Z)\}^4\right] + 2\mathbb{E}\left[\{\tau'(Z)\}^2\{\tau''(Z)\}^2\right] + 3\mathbb{E}\left[\{\tau''(Z)\}^4\right] \right) 3\mathbb{E}\left[\{\tau'(Z)\}^4\right] \right\}^{1/2} \right\}^{1/2}$$

$$= \frac{c_M}{\sqrt{n}} \left\{ 2 \left\{ \mathbb{E}\left[|\tau'(Z)|^2|\tau''(Z)|^2\right] \mathbb{E}\left[|\tau(Z)|^2|\tau'(Z)|^2\right] \right\}^{1/2} \right.$$

$$+ \left\{ \mathbb{E}\left[|\tau'(Z)|^4\right] \mathbb{E}\left[|\tau(Z)|^4\right] \right\}^{1/2}$$

$$+ \left\{ \left( \mathbb{E}\left[|\tau'(Z)|^4\right] + 2\mathbb{E}\left[|\tau'(Z)|^2|\tau''(Z)|^2\right] + 3\mathbb{E}\left[|\tau''(Z)|^4\right] \right) 3\mathbb{E}\left[|\tau'(Z)|^4\right] \right\}^{1/2} \right\}^{1/2},$$

where $Z \sim \mathcal{N}(0,1)$. Now, since $\tau$ is polynomially bounded and the square root is an increasing function,

$$d_M(F,N) \le \frac{c_M}{\sqrt{n}} \left\{ 2 \left\{ \mathbb{E}\left[(a+b|Z|^\gamma)^4\right] \mathbb{E}\left[(a+b|Z|^\gamma)^4\right] \right\}^{1/2} \right.$$

$$+ \left\{ \mathbb{E}\left[(a+b|Z|^\gamma)^4\right] \mathbb{E}\left[(a+b|Z|^\gamma)^4\right] \right\}^{1/2}$$

$$+ \left\{ 18\mathbb{E}\left[(a+b|Z|^\gamma)^4\right] \mathbb{E}\left[(a+b|Z|^\gamma)^4\right] \right\}^{1/2} \right\}^{1/2}$$

$$= \frac{c_M}{\sqrt{n}} \sqrt{3\sqrt{2}+3} \left\{ \mathbb{E}\left[(a+b|Z|^\gamma)^4\right] \right\}^{1/2}$$

$$= \frac{c_M}{\sqrt{n}} \sqrt{3(1+\sqrt{2})} \|a+b|Z|^\gamma\|_{L_4}^2,$$

where $Z \sim \mathcal{N}(0,1)$. $\qquad\square$

The proof of Theorem 3.1 shows how a non-asymptotic approximation of $F$ can be obtained by a direct application of Theorem 2.2. In particular, the estimate equation 8 of the approximation error $d_M(F,N)$ has the optimal rate $n^{-1/2}$ with respect to the 1-Wasserstein distance, the total variation distance and the Kolmogorov-Smirnov distance. As for the constant, it depends on the variance $\mathbb{E}_{Z\sim\mathcal{N}(0,1)}[\tau^2(Z)]$ of $F$. Once the activation function $\tau$ is specified, $\mathbb{E}_{Z\sim\mathcal{N}(0,1)}[\tau^2(Z)]$ can be evaluated by means of an exact or approximate calculation, or a suitable lower bound for it can be provided.

Now, we extend Theorem 3.1 to the more general case of the Gaussian NN equation 1, showing that the problem still reduces to an application of Theorem 2.2. In particular, it is convenient to write equation 1 as follows:

$$F := \frac{1}{n^{1/2}} \sigma_w \sum_{j=1}^{n} w_j \tau(\sigma_w \langle w_j^{(0)}, \boldsymbol{x}\rangle + \sigma_b b_j^{(0)}) + \sigma_b b, \qquad (9)$$

with $w_j^{(0)} = [w_{j,1}^{(0)}, \ldots, w_{j,d}^{(0)}]^T$ and $w_j \overset{d}{=} w_{j,i}^{(0)} \overset{iid}{\sim} \mathcal{N}(0,1)$. We set $\Gamma^2 = \sigma_w^2\|\boldsymbol{x}\|^2 + \sigma_b^2$, and for $n \ge 1$ we consider a collection $(Y_1, \ldots, Y_n)$ of independent standard Gaussian RVs. Then, from equation 9 we can write

$$F \overset{d}{=} \frac{1}{n^{1/2}} \sigma_w \sum_{j=1}^{n} w_j \tau(\Gamma Y_j) + \sigma_b b.$$

As before, straightforward calculations leads to $\mathbb{E}[F] = 0$ and $\mathrm{Var}[F] = \sigma_w^2 \mathbb{E}_{Z\sim\mathcal{N}(0,1)}\left[\tau^2(\Gamma Z)\right] + \sigma_b^2$. As $F$ in equation 9 is a function of independent standard Gaussian RVs, Theorem 2.2 can be applied to approximate $F$ with

a Gaussian RV with the same mean and variance as $F$, quantifying the approximation error. This approximation is stated in the next theorem, whose proof is in Appendix B.

**Theorem 3.2.** *Let $F$ be the NN equation 9 with $\tau \in C^2(\mathbb{R})$ such that $|\tau(x)| \leq a + b|x|^\gamma$ and $\left|\frac{d^l}{dx^l}\tau(x)\right| \leq a + b|x|^\gamma$ for $l = 1, 2$ and some $a, b, \gamma \geq 0$. If $N \sim \mathcal{N}(0, \sigma^2)$ with $\sigma^2 = \sigma_w^2 \mathbb{E}_{Z \sim \mathcal{N}(0,1)}\left[\tau^2(\Gamma Z)\right] + \sigma_b^2$ and $\Gamma = (\sigma_w^2\|x\|^2 + \sigma_b^2)^{1/2}$, then for any $n \geq 1$*

$$d_M(F, N) \leq \frac{c_M \sqrt{\Gamma^2 + \Gamma^4(2 + \sqrt{3(1 + 2\Gamma^2 + 3\Gamma^4)})}\|a + b|\Gamma Z|^\gamma\|_{L^4}^2}{\sqrt{n}}, \tag{10}$$

*where $Z \sim \mathcal{N}(0, 1)$, $M \in \{TV, KS, W_1\}$, with corresponding constants $c_{TV} = 4/\sigma^2$, $c_{KS} = 2/\sigma^2$, $c_{W_1} = \sqrt{8/\sigma^2\pi}$.*

We observe that Theorem 3.1 can be recovered from Theorem 3.2. In particular, the estimate equation 8 of the approximation $d_M(F, N)$ can be recovered from the estimate equation 10 by setting $\sigma_b = 0$, $\sigma_w = 1$ and $x = 1$. As for Theorem 3.1, the constant depends on the variance $\sigma_w^2 \mathbb{E}_{Z \sim \mathcal{N}(0,1)}\left[\tau^2(\Gamma Z)\right] + \sigma_b^2$ of $F$. Therefore, to apply Theorem 3.2 one needs to evaluate the variance of $F$, by means of an exact or approximate calculation, or to provide a suitable lower bound for it, as we have discussed previously.

We conclude by presenting an extension of Theorem 3.2 to a Gaussian NN with $p > 1$ inputs $[x_1, \ldots, x_p]^T$, where $x_i \in \mathbb{R}^d$ for $i = 1, \ldots, p$. In particular, we consider the NN $F := [F_1, \ldots, F_p]^T$ where

$$F_i := \frac{1}{n^{1/2}}\sigma_w \sum_{j=1}^n w_j \tau(\sigma_w \langle w_j^{(0)}, x_i \rangle + \sigma_b b_j^{(0)}) + \sigma_b b, \tag{11}$$

with $w_j^{(0)} = [w_{j,1}^{(0)}, \ldots, w_{j,d}^{(0)}]^T$ and $w_j \stackrel{d}{=} w_{j,i}^{(0)} \stackrel{d}{=} b_j^{(0)} \stackrel{d}{=} b \stackrel{iid}{\sim} \mathcal{N}(0, 1)$. Since the parameter are jointly distributed according to multivariate standard Gaussian distribution, Theorem 2.3 can be applied to approximate $F$ with a Gaussian random vector whose mean and covariance are the same as $F$. The resulting estimate of the approximation error depends on the maximum and the minimum eigenvalues, i.e. $\lambda_1(C)$ and $\lambda_p(C)$ respectively, of the covariance matrix $C$, whose $(i, k)$-th entry is given by

$$\mathbb{E}[F_i F_k] = \sigma_w^2 \mathbb{E}[\tau(Y_i)\tau(Y_k)] + \sigma_b^2, \tag{12}$$

where $Y \sim \mathcal{N}(0, \sigma_w^2 X^T X + \sigma_b^2 \mathbf{1}\mathbf{1}^T)$, with $\mathbf{1}$ being the all-one vector of dimension $p$ and $X$ being the $n \times p$ matrix of the inputs $\{x_i\}_{i \in [p]}$. This approximation is stated in the next theorem.

**Theorem 3.3.** *Let $F = [F_1 \ldots, F_p]^T$ with $F_i$ being the NN equation 11, for $i = 1, \ldots, p$, with $\tau \in C^2(\mathbb{R})$ such that $|\tau(x)| \leq a + b|x|^\gamma$ and $\left|\frac{d^l}{dx^l}\tau(x)\right| \leq a + b|x|^\gamma$ for $l = 1, 2$ and some $a, b, \gamma \geq 0$. Furthermore, let $C$ be the covariance matrix of $F$, whose entries are given in equation 12, and define $\Gamma_i^2 = \sigma_w^2\|x_i\|^2 + \sigma_b^2$ and $\Gamma_{ik} = \sigma_w^2 \sum_{j=1}^d |x_{ij} x_{kj}| + \sigma_b^2$. If $N = [N_1, \cdots N_p]^T \sim \mathcal{N}(0, C)$, then for any $n \geq 1$*

$$d_{W_1}(F, N) \leq 2\sigma_w^2 \tilde{K} \frac{\lambda_1(C)}{\lambda_p(C)}\sqrt{\frac{p}{n}}, \tag{13}$$

*where $\lambda_1(C)$ and $\lambda_p(C)$ are the maximum and the minimum eigenvalues of $C$, respectively, and where*

$$\tilde{K} = \left\{\sum_{i,k=1}^p (\Gamma_i^2 + \sqrt{3(1 + 2\Gamma_i^2 + 3\Gamma_i^4)}\Gamma_{ik}^2 + 2\Gamma_i^2\Gamma_{ik})\|a + b|\Gamma_i Z|^\gamma\|_{L^4}^2\|a + b|\Gamma_k Z|^\gamma\|_{L^4}^2\right\}^{1/2},$$

*with $Z \sim \mathcal{N}(0, 1)$.*

*Proof.* The proof is based on Theorem 2.3. Recall that

$$F_i := \frac{1}{n^{1/2}}\sigma_w \sum_{j=1}^n w_j \tau(\sigma_w \langle w_j^{(0)}, x_i \rangle + \sigma_b b_j^{(0)}) + \sigma_b b.$$

Since $F_1, \ldots, F_p$ are functions of the iid standard normal random variables $\{w_j, w_{jl}^{(0)}, b_j^{(0)}, b : j = 1, \ldots, n, l = 1, \ldots, d\}$, then we can apply Theorem 2.3 to the random vector $F = [F_1, \cdots, F_p]^T$. The upper bound in equation 6 depends on the first and second derivatives of the $F_i$'s with respect to all their arguments. However, the derivatives with respect to $b$ give no contributions, since, for every $i = 1, \ldots, p$, $\nabla_{b,.}^2 F_i$ is the zero vector. Moreover, the terms $w_j \tau(\sigma_w \langle w_j^{(0)}, x_i \rangle + \sigma_b b_j^{(0)})$ are iid, across $j$, and give the same contribution to the upper bound. Hence, we can write that

$$d_{W_1}(F, N) \leq 2\sigma_w^2 \sqrt{\frac{p}{n}} \|C^{-1}\|_2 \|C\|_2 \sqrt{\sum_{i,k=1}^p D_{ik}},$$

where

$$D_{ik} = \sum_{l,m} \left\{ \mathbb{E}\left[ (\langle \nabla_{l,.}^2 \tilde{F}_i, \nabla_{m,.}^2 \tilde{F}_i \rangle)^2 \right] \right\}^{1/2} \left\{ \mathbb{E}\left[ (\nabla_l \tilde{F}_k \nabla_m \tilde{F}_k)^2 \right] \right\}^{1/2},$$

where

$$[\tilde{F}_1, \ldots, \tilde{F}_p] \stackrel{d}{=} [w_j \tau(\sigma_w \langle w_j^{(0)}, x_1 \rangle + \sigma_b b_j^{(0)}), \ldots, w_j \tau(\sigma_w \langle w_j^{(0)}, x_p \rangle + \sigma_b b_j^{(0)})],$$

and $\nabla_l, \nabla_m, \nabla_{l,.}^2$ and $\nabla_{m,.}^2$ denote the derivatives with respect to all the arguments. We can represent $\tilde{F}_i$ as

$$\tilde{F}_i = w \cdot \tau(Y_i),$$

where $Y_i := \langle \tilde{w}^{(0)}, \tilde{x}_i \rangle = \sum_{s=1}^d \tilde{w}_s^{(0)} \tilde{x}_{is}$, with $\tilde{x}_i := [\sigma_w x_i^T, \sigma_b]^T$, $\tilde{w}^{(0)} := [w^{(0)T}, b^{(0)}]^T$, and $w, \tilde{w}_1^{(0)}, \ldots, \tilde{w}_d^{(0)}, b^{(0)}$ iid standard normal random variables. The gradient and the Hessian of $\tilde{F}$ with respect to the parameters $w$ and $\tilde{w}_s^{(0)}$ are

$$\begin{cases} \frac{\partial \tilde{F}_i}{\partial w} = \tau(Y_i) \\[2mm] \frac{\partial \tilde{F}_i}{\partial w_s^{(0)}} = w\tau'(Y_i)\tilde{x}_{is} \\[2mm] \nabla_{w,w}^2 \tilde{F}_i = 0 \\[2mm] \nabla_{w,\tilde{w}_s^{(0)}}^2 \tilde{F}_i = \tau'(Y_i)\tilde{x}_{is} \\[2mm] \nabla_{\tilde{w}_s^{(0)},\tilde{w}_t^{(0)}}^2 \tilde{F}_i = w\tau''(Y_i)\tilde{x}_{is}\tilde{x}_{it}. \end{cases}$$

This implies that

$$D_{ik} = \left\{ \mathbb{E}\left[ \left( \sum_{s=1}^d \nabla_{w,\tilde{w}_s^{(0)}}^2 \tilde{F}_i \cdot \nabla_{w,\tilde{w}_s^{(0)}}^2 \tilde{F}_i \right)^2 \right] \right\}^{1/2} \left\{ \mathbb{E}\left[ \left( \frac{\partial \tilde{F}_k}{\partial w} \cdot \frac{\partial \tilde{F}_k}{\partial w} \right)^2 \right] \right\}^{1/2}$$

$$+ \sum_{j,j'=1}^d \left\{ \mathbb{E}\left[ \left( \nabla_{w,\tilde{w}_j^{(0)}}^2 \tilde{F}_i \cdot \nabla_{w,\tilde{w}_{j'}^{(0)}}^2 \tilde{F}_i + \sum_{s=1}^d \nabla_{\tilde{w}_j^{(0)},\tilde{w}_s^{(0)}}^2 \tilde{F}_i \cdot \nabla_{\tilde{w}_{j'}^{(0)},\tilde{w}_s^{(0)}}^2 \tilde{F}_i \right)^2 \right] \right\}^{1/2}$$

$$\times \left\{ \mathbb{E}\left[ \left( \frac{\partial \tilde{F}_k}{\partial \tilde{w}_j^{(0)}} \cdot \frac{\partial \tilde{F}_k}{\partial \tilde{w}_{j'}^{(0)}} \right)^2 \right] \right\}^{1/2}$$

$$+ 2\sum_{j=1}^d \left\{ \mathbb{E}\left[ \left( \sum_{s=1}^d \nabla_{w,\tilde{w}_s^{(0)}}^2 \tilde{F}_i \cdot \nabla_{\tilde{w}_j^{(0)},\tilde{w}_s^{(0)}}^2 \tilde{F}_i \right)^2 \right] \right\}^{1/2} \left\{ \mathbb{E}\left[ \left( \frac{\partial \tilde{F}_k}{\partial w} \cdot \frac{\partial \tilde{F}_k}{\partial \tilde{w}_j^{(0)}} \right)^2 \right] \right\}^{1/2}$$

$$= \left\{ \mathbb{E}\left[ \left( \sum_{s=1}^d \tau'(Y_i)^2 \tilde{x}_{is}^2 \right)^2 \right] \right\}^{1/2} \left\{ \mathbb{E}\left[ (\tau(Y_k))^4 \right] \right\}^{1/2}$$

$$+ \sum_{j,j'=1}^{d} \left\{ \mathbb{E}\left[ \left( \tau'(Y_i)^2 \tilde{x}_{ij} \tilde{x}_{ij'} + \sum_{s=1}^{d} w^2 \tau''(Y_i)^2 \tilde{x}_{ij} \tilde{x}_{ij'} \tilde{x}_{is}^2 \right)^2 \right] \right\}^{1/2} \left\{ \mathbb{E}\left[ \left( w^2 \tau'(Y_k)^2 \tilde{x}_{kj} \tilde{x}_{kj'} \right)^2 \right] \right\}^{1/2}$$

$$+ 2\sum_{j=1}^{d} \left\{ \mathbb{E}\left[ \left( \sum_{s=1}^{d} \tau'(Y_i) \tilde{x}_{is} w \tau''(Y_i) \tilde{x}_{ij} \tilde{x}_{is} \right)^2 \right] \right\}^{1/2} \left\{ \mathbb{E}\left[ \left( \tau(Y_k) w \tau'(Y_k) \tilde{x}_{kj} \right)^2 \right] \right\}^{1/2}$$

$$= ||\tilde{\boldsymbol{x}}_i||^2 \|\tau'(Y_i)\|_{L_4}^2 \|\tau(Y_k)\|_{L_4}^2$$

$$+ \sum_{j,j'=1}^{d} |\tilde{x}_{ij} \tilde{x}_{ij'}| \left\{ \mathbb{E}\left[ \left( \tau'(Y_i)^2 + w^2 \tau''(Y_i)^2 ||\tilde{\boldsymbol{x}}_i||^2 \right)^2 \right] \right\}^{1/2} \sqrt{3} |\tilde{x}_{kj} \tilde{x}_{kj'}| \|\tau'(Y_k)\|_{L^4}^2$$

$$+ 2\sum_{j=1}^{d} |\tilde{x}_{ij}||\tilde{x}_{kj}| \|\tilde{\boldsymbol{x}}_i\|^2 \left\{ \mathbb{E}\left[ \left( \tau'(Y_i) \tau''(Y_i) \right)^2 \right] \right\}^{1/2} \left\{ \mathbb{E}\left[ \left( \tau(Y_k) \tau'(Y_k) \right)^2 \right] \right\}^{1/2}$$

$$= ||\tilde{\boldsymbol{x}}_i||^2 \|\tau'(Y_i)\|_{L_4}^2 \|\tau(Y_k)\|_{L_4}^2 + \sum_{j,j'=1}^{d} \sqrt{3} |\tilde{x}_{kj} \tilde{x}_{kj'}||\tilde{x}_{ij} \tilde{x}_{ij'}| \|\tau'(Y_k)\|_{L^4}^2$$

$$\times \left\{ \|\tau'(Y_i)\|_{L_4}^4 + 3\|\tilde{\boldsymbol{x}}_i\|^4 \left\|\tau''(Y_i)\right\|_{L_4}^4 + 2\|\tilde{\boldsymbol{x}}_i\|^2 \left\|\tau'(Y_i) \tau''(Y_i)\right\|_{L_2}^2 \right\}^{1/2}$$

$$+ 2\sum_{j=1}^{d} |\tilde{x}_{ij}||\tilde{x}_{kj}| \|\tilde{\boldsymbol{x}}_i\|^2 \left\|\tau'(Y_i) \tau''(Y_i)\right\|_{L_2} \left\|\tau(Y_k) \tau'(Y_k)\right\|_{L_2}$$

$$= ||\tilde{\boldsymbol{x}}_i||^2 \|\tau'(Y_i)\|_{L_4}^2 \|\tau(Y_k)\|_{L_4}^2 + \sqrt{3} \|\tau'(Y_k)\|_{L^4}^2 \left( \sum_{j=1}^{d} |\tilde{x}_{ij} \tilde{x}_{kj}| \right)^2$$

$$\times \left\{ \|\tau'(Y_i)\|_{L_4}^4 + 3\|\tilde{\boldsymbol{x}}_i\|^4 \left\|\tau''(Y_i)\right\|_{L_4}^4 + 2\|\tilde{\boldsymbol{x}}_i\|^2 \left\|\tau'(Y_i) \tau''(Y_i)\right\|_{L_2}^2 \right\}^{1/2}$$

$$+ 2\|\tilde{\boldsymbol{x}}_i\|^2 \left\|\tau'(Y_i) \tau''(Y_i)\right\|_{L_2} \left\|\tau(Y_k) \tau'(Y_k)\right\|_{L_2} \left( \sum_{j=1}^{d} |\tilde{x}_{ij}||\tilde{x}_{kj}| \right)$$

$$\overset{\text{Holder ineq.}}{\leq} ||\tilde{\boldsymbol{x}}_i||^2 \|\tau'(Y_i)\|_{L_4}^2 \|\tau(Y_k)\|_{L_4}^2 + \sqrt{3} \|\tau'(Y_k)\|_{L^4}^2 \left( \sum_{j=1}^{d} |\tilde{x}_{ij} \tilde{x}_{kj}| \right)^2$$

$$\times \left\{ \|\tau'(Y_i)\|_{L_4}^4 + 3\|\tilde{\boldsymbol{x}}_i\|^4 \left\|\tau''(Y_i)\right\|_{L_4}^4 + 2\|\tilde{\boldsymbol{x}}_i\|^2 \left\|\tau'(Y_i)\right\|_{L_4}^2 \left\|\tau''(Y_i)\right\|_{L_4}^2 \right\}^{1/2}$$

$$+ 2\|\tilde{\boldsymbol{x}}_i\|^2 \left\|\tau'(Y_i)\right\|_{L_4} \left\|\tau''(Y_i)\right\|_{L_4} \|\tau(Y_k)\|_{L_4} \left\|\tau'(Y_k)\right\|_{L_4} \left( \sum_{j=1}^{d} |\tilde{x}_{ij}||\tilde{x}_{kj}| \right)$$

$$\overset{\text{polynom. bounded}}{\leq} ||\tilde{\boldsymbol{x}}_i||^2 \|a + b|Y_i|^{\gamma}\|_{L_4}^2 \|a + b|Y_k|^{\gamma}\|_{L_4}^2$$

$$+ \sqrt{3} \left\{ (1 + 2\|\tilde{\boldsymbol{x}}_i\|^2 + 3\|\tilde{\boldsymbol{x}}_i\|^4)\|a + b|Y_i|^{\gamma}\|_{L_4}^4 \right\}^{1/2} \|a + b|Y_k|^{\gamma}\|_{L^4}^2 \left( \sum_{j=1}^{d} |\tilde{x}_{ij} \tilde{x}_{kj}| \right)^2$$

$$+ 2\|\tilde{\boldsymbol{x}}_i\|^2 \|a + b|Y_i|^{\gamma}\|_{L_4}^2 \|a + b|Y_k|^{\gamma}\|_{L_4}^2 \left( \sum_{j=1}^{d} |\tilde{x}_{ij} \tilde{x}_{kj}| \right)$$

$$= \left\{ ||\tilde{\boldsymbol{x}}_i||^2 + \sqrt{3(1 + 2||\tilde{\boldsymbol{x}}_i||^2 + 3||\tilde{\boldsymbol{x}}_i||^4)} \left( \sum_{j=1}^d |\tilde{x}_{ij}\tilde{x}_{kj}| \right)^2 + 2||\tilde{\boldsymbol{x}}_i||^2 \left( \sum_{j=1}^d |\tilde{x}_{ij}\tilde{x}_{kj}| \right) \right\}$$
$$\times \|a + b|Y_i|^\gamma\|_{L_4}^2 \|a + b|Y_k|^\gamma\|_{L_4}^2.$$

Now, traducing everything back to the original variables $\{\boldsymbol{x}_i\}_{i \in [d]}$, we have that

$$\begin{cases} \sum_{j=1}^d |\tilde{x}_{ij}||\tilde{x}_{kj}| = \sigma_w^2 \sum_{j=1}^d |x_{ij}||x_{kj}| + \sigma_b^2 =: \Gamma_{ik} \\ \\ ||\tilde{\boldsymbol{x}}_i||^2 = \sigma_w^2 ||\boldsymbol{x}_i||^2 + \sigma_b^2 =: \Gamma_i^2. \end{cases}$$

Hence,

$$D_{ik} \leq (\Gamma_i^2 + \sqrt{3(1 + 2\Gamma_i^2 + 3\Gamma_i^4)}\Gamma_{ik}^2 + 2\Gamma_i^2\Gamma_{ik})\|a + b|Y_i|^\gamma\|_{L_4}^2 \|a + b|Y_k|^\gamma\|_{L_4}^2,$$

with $Y \sim \mathcal{N}(0, \sigma_b^2 X^T X + \sigma_b^2 \mathbf{1}\mathbf{1}^T)$. Summing over all possible $i, k = 1, \dots, p$ and taking the square root leads to

$$d_{W_1}(F, N) \leq 2\sigma_w^2 \frac{\lambda_1(C)}{\lambda_p(C)} \sqrt{\frac{p}{n}} \tilde{K},$$

with

$$\tilde{K} = \left\{ \sum_{i,k=1}^p (\Gamma_i^2 + \sqrt{3(1 + 2\Gamma_i^2 + 3\Gamma_i^4)}\Gamma_{ik}^2 + 2\Gamma_i^2\Gamma_{ik})\|a + b|Y_i|^\gamma\|_{L^4}^2 \|a + b|Y_k|^\gamma\|_{L^4}^2 \right\}^{1/2}$$
$$= \left\{ \sum_{i,k=1}^p (\Gamma_i^2 + \sqrt{3(1 + 2\Gamma_i^2 + 3\Gamma_i^4)}\Gamma_{ik}^2 + 2\Gamma_i^2\Gamma_{ik})\|a + b|\Gamma_i Z|^\gamma\|_{L^4}^2 \|a + b|\Gamma_k Z|^\gamma\|_{L^4}^2 \right\}^{1/2},$$

with $Z \sim \mathcal{N}(0, 1)$, which concludes the proof. $\qquad\square$

The estimate equation 13 of the approximation error $d_{W_1}(F, N)$ depends on the spectral norms of the covariance matrix $C$ and the precision matrix $C^{-1}$. Such spectral norms must be computed explicitly for the specific activation $\tau$ in use, or at least bounded from above, in order to apply Theorem 3.3. This boils down to finding the greatest eigenvalue $\lambda_1$ and the smallest eigenvalue $\lambda_p$ of the matrix $C$, which can be done for a broad class of activations with classical optimization techniques, or at least bounding $\lambda_1$ from above and $\lambda_p$ from below (Diaconis & Stroock, 1991; Guattery et al., 1999).

## 4 Numerical illustrations

In this section, we present a brief simulation study for two specific choices of the activation function: i) $\tau(x) = \tanh x$, which is polynomially bounded with parameters $a = 1$ and $b = 0$; ii) $\tau(x) = x^3$, which is polynomially bounded with parameters $a = 6$, $b = 1$ and $\gamma = 3$. Each of the plots below is obtained as follows: for a fixed width of $n = k^3$, with $k \in \{1, \cdots, 16\}$, we simulate 5000 points from a SLNN as in Theorem 3.1 to produce an estimate of the distance between the NN and a Gaussian RV with mean 0 and variance $\sigma^2$, which is estimated by means of a Monte-Carlo approach. Estimates of the KS and TV distance are produced by means of the functions *KolmogorovDist* and *TotVarDist* from the package **distrEx** by Ruckdeschel et al. (2006) while those of the 1-Wasserstein distance using the function *wasserstein1d* from the package **transport** by Schuhmacher et al. (2022). We repeat this procedure 500 times for every fixed $n = k^3$ with $k \in \{1, \cdots, 16\}$, compute the sample mean (black dots) and the 2.5-th and the 97.5-th sample percentiles (red dashed lines), and compare these estimates with the theoretical bound given by Theorem 3.1 (blue line).

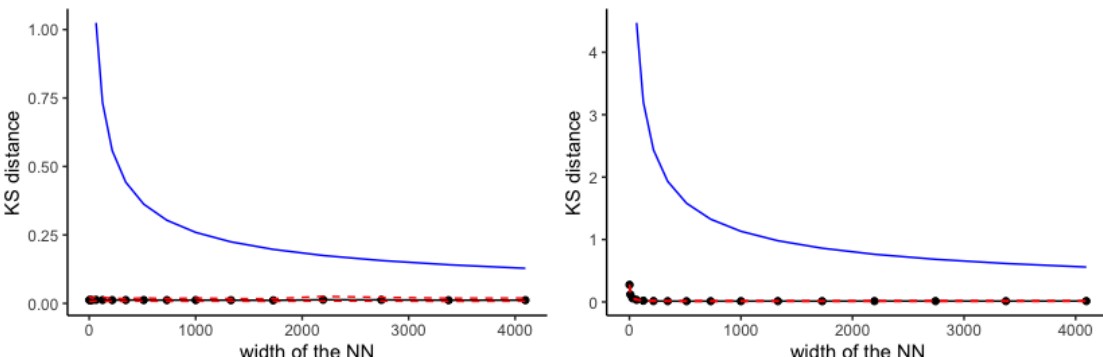

Figure 1: Estimates of the Kolmogorov-Smirnov distance for a Shallow NN of varying width $n = k^3$, $k \in \{1, \cdots, 16\}$, with $\tau(x) = \tanh x$ (left) and $\tau(x) = x^3$ (right). The blue line is the theoretical bound of Theorem 3.1, the black dots are sample means of the Monte-Carlo sample, while the red-dashed lines represent a 95% sample confidence interval.

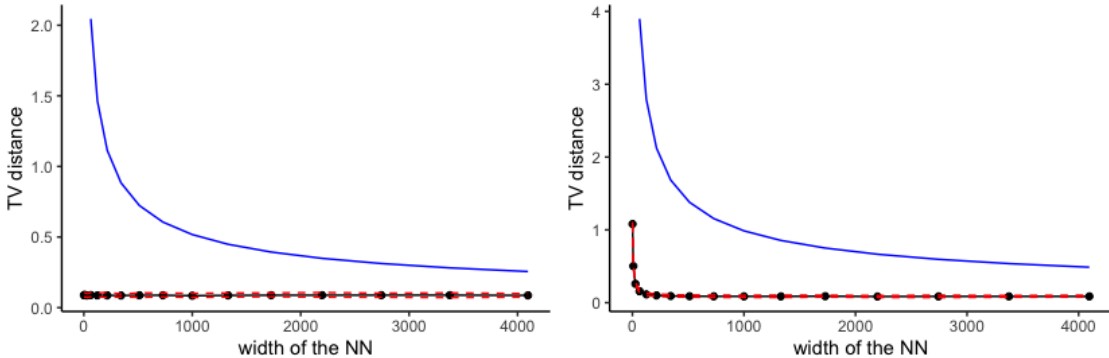

Figure 2: Estimates of the Total Variation distance for a Shallow NN of varying width $n = k^3$, $k \in \{1, \cdots, 16\}$, with $\tau(x) = \tanh x$ (left) and $\tau(x) = x^3$ (right).

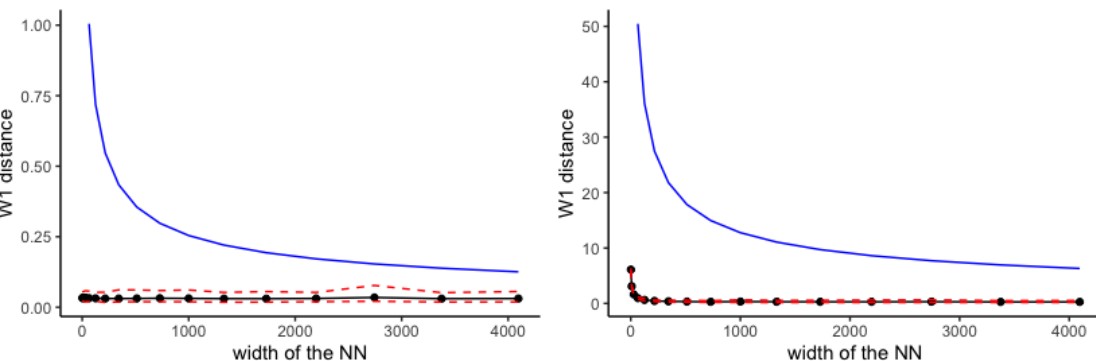

Figure 3: Estimates of the 1-Wasserstein distance for a Shallow NN of varying width $n = k^3$, $k \in \{1, \cdots, 16\}$, with $\tau(x) = \tanh x$ (left) and $\tau(x) = x^3$ (right).

The plots confirm that the distance between a shallow NN and an arbitrary Gaussian RV, with the same mean and variance, is asymptotically bounded from above by $n^{-1/2}$ and that the approximation gets better and better as $n \to \infty$. This is evident in the case $\tau(x) = x^3$, where there is a clear decay between $n = 1$ and $n = 1000$. This behaviour does not show up for $\tau(x) = \tanh x$, since $\tanh x \sim x$, for $x \to 0$, and Gaussian RVs are more likely to attain values in a neighbourhood of zero.

## 5 Discussion

We introduced some non-asymptotic Gaussian approximations of Gaussian NNs, quantifying the approximation error with respect to the 1-Wasserstein distance, the total variation distance and the Kolmogorov-Smirnov distance. As a novelty, our work relies on the use of second-order Poincaré inequalities, which lead to estimates of the approximation error with optimal rate and tight constants. This is the first work to make use of second-order Poincaré inequalities for non-asymptotic Gaussian approximations of Gaussian NNs. For a Gaussian NN with a single input, the estimate in Theorem 3.2 requires to evaluate or estimate $\sigma_w^2 \mathbb{E}_{Z \sim \mathcal{N}(0,1)} \left[ \tau^2 \left( \Gamma Z \right) \right] + \sigma_b^2$, whereas for a Gaussian NN with $p > 1$ inputs, the estimate in Theorem 3.3 requires to evaluate or estimate $\left\| C^{-1} \right\|_2 \left\| C \right\|_2$. Our approach based on second-order Poincaré inequalities remains valid in the more general setting of deep Gaussian NNs. Both Theorem 3.2 and Theorem 3.3 can be extended to deep Gaussian NNs, at the cost of more involved algebraic calculations, as well as more involved estimates of the approximation errors. For instance, for an input $\boldsymbol{x}$ one may consider a deep Gaussian NN with $L \geq 1$ layers, i.e.

$$
\begin{cases}
g_j^{(1)}(\boldsymbol{x}) = \sigma_w \langle w_j^{(0)}, \boldsymbol{x} \rangle_{\mathbb{R}^d} + \sigma_b b_j^{(0)} \\
g_j^{(l)}(\boldsymbol{x}) = \sigma_b b_j^{(l-1)} + \sigma_w n^{-1/2} \sum_{i=1}^n w_{j,i}^{(l-1)} \tau(g_i^{(l-1)}(\boldsymbol{x})), \quad \forall l = 2, ..., L \\
f_{\boldsymbol{x}}(n)[\tau, \alpha] = g_1^{(L+1)}(\boldsymbol{x}) = \sigma_b b + \sigma_w n^{-1/2} \sum_{j=1}^n w_j \tau(g_j^{(L)}(\boldsymbol{x}))
\end{cases}
\tag{14}
$$

with

$$
\begin{cases}
w_i^{(0)} = [w_{i,1}^{(0)}, w_{i,2}^{(0)}, \ldots w_{i,d}^{(0)}] & \in \mathbb{R}^d \\
w_i^{(l)} := [w_{i,1}^{(l)}, w_{i,2}^{(l)}, \ldots w_{i,n}^{(l)}] & \in \mathbb{R}^n \\
w = [w_1, w_2, \ldots w_n] & \in \mathbb{R}^n \\
w_{i,j}^{(0)}, w_{i,j}^{(l)}, w_i, b_i^{(l)}, b & \in \mathbb{R} \\
w_{i,j}^{(0)} \overset{d}{=} w_{i,j}^{(l)} \overset{d}{=} w_j \overset{d}{=} b_i^{(l)} \overset{d}{=} b \overset{iid}{\sim} \mathcal{N}(0,1)
\end{cases}
\tag{15}
$$

and apply Theorem 2.2 to $F := g_1^{(L+1)}(\boldsymbol{x})$ as defined in equation 14 and equation 15. Such an application implies to deals with complicated expressions of the gradient and the Hessian that, however, is a purely algebraic problem.

Related to the choice of the activation, one can also try to relax the hypothesis of polynomially boundedness and use a whatever $\tau \in C^2(\mathbb{R})$. There is nothing wrong in doing it, as Corollary 2.2 and 2.3 still apply, with the only difference that the bound would be less explicit than the one we found here. Furthermore, one could also think about relaxing the $C^2(\mathbb{R})$ hypothesis to include $C^1(\mathbb{R})$ or just continuous activations, like the famous ReLU function (i.e. $\mathrm{ReLU}(x) = \max\{0, x\}$) which is excluded from our analysis. Some results in this direction can be found in Eldan et al. (2021), though using Rademacher weights for the hidden layer. In this regard, we try to derive a specific bound for the ReLU function applying Corollary 2.2 to a sequence of smooth approximating functions and then passing to the limit. In particular, we approximated the ReLU function with $G(m, x) := m^{-1} \log(1 + e^{mx})$ for $m \geq 1$ and applied Theorem 2.2 to a generic $G(m, x)$ using the 1-Wasserstein distance and obtained a bound dependent on $m$. Then, the idea would have been to take the limit of this bound for $m \to \infty$ and hopefully obtain a non-trivial bound, but that was not the case as the limit exploded. The same outcome was found using the SAU approximating sequence, i.e.

$$
H(m, x) := \frac{1}{m\sqrt{2\pi}} \exp\left\{ -\frac{1}{2} m^2 x^2 \right\} + \frac{x}{2} + \frac{x}{2} \mathrm{erf}\left\{ \frac{mx}{\sqrt{2}} \right\},
$$

where $\mathrm{erf}\left(\cdot\right)$ denotes the error function. This fact probably indicates the impossibility to apply the results of Vidotto (2020) in the context of continuous activation functions as the ReLU function, and the necessity to come up with new results on second-order Poincaré inequalities to fill this gap. These results would not be trivial aft all, since Theorem A.2 needs each $F_1, \ldots, F_d$ to be in $\mathbb{D}^{2,4}$, and so two degrees of smoothness are required. This is not "the fault" of Vidotto (2020), but it is due to the intrinsic character of the equation $f''(x) - x f'(x) = h(x) - Eh(Z)$ with $Z \sim \mathcal{N}(0,1)$ in dimension $p \geq 2$.

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

# A  Second-order Poincaré inequality for functionals of Gaussian fields

We present a brief overview of the main results of Vidotto (2020), of which Theorem 2.2 and Theorem 2.3 are special cases for RVs in $\mathbb{R}^d$. The main results of Vidotto (2020) improve on previous results of Nourdin et al. (2009), and such an improvement is obtained by using the Mehler representation of the Ornstein–Uhlenbeck semigroup, which was exploited in Last et al. (2016) to obtain second-order Poincaré inequalities for Poisson functionals. According to the Mehler formula, if $F \in L^1$, $X'$ is an independent copy of a RV $X$, with $X$ and $X'$ being defined on the product probability space $(\Omega \times \Omega', \mathcal{F} \otimes \mathcal{F}', \mathbb{P} \times \mathbb{P}')$, and $P_t$ is the infinitesimal generator of the Ornstein–Uhlenbeck process then

$$P_t F = E\left[ f\left( e^{-t} X + \sqrt{1 - e^{-2t}} X' \right) \mid X \right], \quad t \geq 0.$$

Before stating Vidotto (2020, Theorem 2.1), it is useful to introduce some notation and definitions from Gaussian analysis and Malliavin calculus. We recall that an isonormal Gaussian process $X = \{X(h) : h \in H\}$ over $H = L^2(A, \mathcal{B}(A), \mu)$, where $(A, \mathcal{B}(A))$ is a Polish space endowed with its Borel $\sigma$-field and $\mu$ is a positive, $\sigma$-finite and non-atomic measure, is a centered Gaussian family defined on $(\Omega, \mathcal{F}, \mathbb{P})$ such that $E[X(h)X(g)] = \langle g, h \rangle_H$ for every $h, g \in H$. We denote by $L^2(\Omega; H)$ the set of $H$-valued random variables $Y$ satisfying $\mathbb{E}[\|Y\|_H^2] < \infty$. Furthermore, if $\mathcal{S}$ denotes the set of RVs of the form

$$F = f\left( X(\phi_1), \ldots, X(\phi_m) \right),$$

where $f : \mathbb{R}^m \to \mathbb{R}$ is a $C^\infty$-function such that $f$ and its partial derivatives have at most polynomial growth at infinity, and $\phi_i \in H$, for $i = 1, \ldots, m$, the Malliavin derivative of $F$ is the element of $L^2(\Omega; H)$ defined by

$$DF = \sum_{i=1}^m \frac{\partial f}{\partial x_i} \left( X(\phi_1), \ldots, X(\phi_m) \right) \phi_i.$$

Moreover, in analogy with $DF$, the second Malliavin derivative of $F$ is the element of $L^2(\Omega; H^{\odot})$ defined by

$$D^2 F = \sum_{i,j=1}^m \frac{\partial^2 f}{\partial x_i \partial x_j} \left( X(\phi_1), \ldots, X(\phi_m) \right) \phi_i \phi_j,$$

where $H^{\odot 2}$ is the second symmetric tensor power of $H$, so that $H^{\odot 2} = L_s^2\left( A^2, \mathcal{B}\left( A^2 \right), \mu^2 \right)$ is the subspace of $L^2\left( A^2, \mathcal{B}\left( A^2 \right), \mu^2 \right)$ whose elements are a.e. symmetric. Let us also define the Sobolev spaces $\mathbb{D}^{\alpha, p}, p \geq 1, \alpha = 1, 2$, which are defined as the closure of $\mathcal{S}$ with respect to the norms

$$\|F\|_{\mathbb{D}^{\alpha, p}} = \left( E\left[ |F|^p \right] + E\left[ \|DF\|_H^p + E\left[ \left\| D^2 F \right\|_{H^{\otimes 2}}^p \right] \mathbb{1}_{\{\alpha = 2\}} \right] \right)^{1/p}.$$

In particular, the Sobolev space $\mathbb{D}^{\alpha,p}$ is typically referred to as the domain of $D^\alpha$ in $L^p(\Omega)$. Finally, for every $1 \leq m \leq n$, every $r = 1, \ldots, m$, every $f \in L^2(A^m, \mathscr{B}(A^m), \mu^m)$ and every $g \in L^2(A^n, \mathscr{B}(A^n), \mu^n)$, the $r$-th contraction $f \otimes_r g : A^{n+m-2r} \to \mathbb{R}$ is defined to be the following function:

$$f \otimes_r g\, (y_1, \ldots, y_{n+m-2r}) = \int_{A^r} f(x_1, \ldots, x_r, y_1, \ldots, y_{m-r})$$
$$\times g(x_1, \ldots, x_r, y_{m-r+1}, \ldots, y_{m+n-2r})\, \mathrm{d}\mu(x_1) \cdots \mathrm{d}\mu(x_r).$$

Now, we can state Vidotto (2020, Theorem 2.1), which provides a second-order Poincaré inequality for a suitable class of functionals of Gaussian fields. For RVs in $\mathbb{R}^d$, the next theorem reduces to Theorem 2.2.

**Theorem A.1** (Vidotto (2020), Theorem 2.1). *Let $F \in \mathbb{D}^{2,4}$ be such that $E[F] = 0$ and $E[F^2] = \sigma^2$, and let $N \sim \mathcal{N}(0, \sigma^2)$; then,*

$$d_M(F, N) \leq c_M \left( \int_{A \times A} \left\{ E\left[ \left( (D^2 F \otimes_1 D^2 F)(x, y) \right)^2 \right] \right\}^{1/2} \right.$$
$$\left. \times \left\{ E\left[ (DF(x)DF(y))^2 \right] \right\}^{1/2} \, \mathrm{d}\mu(x)\mathrm{d}\mu(y) \right)^{1/2}$$

*where $M \in \{TV, KS, W_1\}$ and $c_{TV} = \frac{4}{\sigma^2}, c_{KS} = \frac{2}{\sigma^2}, c_{W_1} = \sqrt{\frac{8}{\sigma^2 \pi}}$.*

The novelty of Theorem A.1 lies in the fact that the upper bound is directly computable, making the approach of Vidotto (2020) very appealing for concrete applications of the Gaussian approximation. In particular, Theorem A.1 improves over previous results of Chatterjee (2009) and Nourdin et al. (2009). Now, we can state Vidotto (2020, Theorem 2.3), which provides a generalization of Theorem A.1 to multidimensional functionals. For RVs in $\mathbb{R}^d$, the next theorem reduces to Theorem 2.3.

**Theorem A.2** (Vidotto (2020), Theorem 2.3). *Let $F = (F_1, \ldots, F_p)$, where, for each $i = 1, \ldots, p, F_i \in \mathbb{D}^{2,4}$ is such that $E[F_i] = 0$ and $E[F_i F_j] = c_{ij}$, with $C = \{c_{ij}\}_{i,j=1,\ldots,p}$ a symmetric and positive definite matrix. Let $N \sim \mathcal{N}(0, C)$, then we have that $d_{W_1}(F, N) \leq 2\sqrt{p} \left\| C^{-1} \right\|_{op} \| C \|_{op} \times$*

$$\sqrt{\sum_{i,k=1}^{p} \int_{A \times A} \left\{ E\left[ \left( (D^2 F_i \otimes_1 D^2 F_i)(x, y) \right)^2 \right] \right\}^{1/2} \left\{ E\left[ (DF_k(x)DF_k(y))^2 \right] \right\}^{1/2} \, \mathrm{d}\mu(x)\mathrm{d}\mu(y)}.$$

## B  Proof of Theorem 3.2

As stated in the main body, we will make use of the fact that

$$F \stackrel{d}{=} \tilde{F} := n^{-1/2} \sigma_w \sum_{j=1}^{n} w_j \tau(\Gamma \cdot Y_j) + \sigma_b \cdot b,$$

where $\Gamma = \sigma_w^2 \|x\|^2 + \sigma_b^2$. First, it is easy to see that $\mathbb{E}[F] = 0$ and that

$$\sigma^2 = \mathrm{Var}[F] = \mathrm{Var}[\tilde{F}] = \sigma_w^2 \mathbb{E}_{Z \sim \mathcal{N}(0,1)}\left[ \tau^2(\Gamma Z) \right] + \sigma_b^2.$$

Then we have that $d_M(F, N) = d_M(\tilde{F}, N)$, where $N \sim \mathcal{N}(0, \sigma^2)$, hence it is enough to apply Theorem 2.2 to $\tilde{F}$. To this aim, we compute again the gradient and the Hessian of $\tilde{F}$, noticing that the only difference with the Shallow case lies in the presence of an extra factor $\sigma_w$ in front of the sum, an extra factor of $\Gamma$ inside the activation and the bias term

$\sigma_b^2 b$:

$$
\begin{cases}
\frac{\partial \tilde{F}}{\partial b} = \sigma_b \\[2mm]
\frac{\partial \tilde{F}}{\partial w_j} = n^{-1/2} \sigma_w \cdot \tau\left(\Gamma Y_j\right) \\[2mm]
\frac{\partial \tilde{F}}{\partial Y_j} = n^{-1/2} \sigma_w \Gamma \cdot w_j \cdot \tau'\left(\Gamma Y_j\right) \\[2mm]
\nabla_{b,\cdot}^2 \tilde{F} = 0 \\[2mm]
\nabla_{w_j,w_i}^2 \tilde{F} = 0 \\[2mm]
\nabla_{w_j,Y_i}^2 \tilde{F} = n^{-1/2} \sigma_w \Gamma \cdot \tau'\left(\Gamma Y_j\right) \delta_{ij} \\[2mm]
\nabla_{Y_j,Y_i}^2 \tilde{F} = n^{-1/2} \sigma_w \Gamma^2 \cdot w_j \cdot \tau''\left(\Gamma Y_j\right) \delta_{ij}
\end{cases}
$$

It is interesting to notice that since the row of the Hessian corresponding to the bias term $b$ contains all zeros, then the bound given by Corollary 2.2 is exactly the same as the one at the beginning of the proof of Theorem 3.1, with the only difference that now the expectations depend also on $\Gamma$ and $\sigma_w$. More precisely, we have that

$$
d_M\left(F, N\right) = d_M\left(\tilde{F}, N\right) \leq
$$

$$
\leq c_M \left\{ \sum_{j=1}^{n} 2 \left\{ \mathbb{E}\left[\left(\langle \nabla_{w_j,\cdot}^2 \tilde{F}, \nabla_{Y_j,\cdot}^2 \tilde{F} \rangle\right)^2\right] \cdot \mathbb{E}\left[\left(\frac{\partial \tilde{F}}{\partial w_j} \cdot \frac{\partial \tilde{F}}{\partial Y_j}\right)^2\right] \right\}^{1/2} \right.
$$

$$
+ \left\{ \mathbb{E}\left[\left(\langle \nabla_{w_j,\cdot}^2 \tilde{F}, \nabla_{w_j,\cdot}^2 \tilde{F} \rangle\right)^2\right] \cdot \mathbb{E}\left[\left(\frac{\partial \tilde{F}}{\partial w_j} \cdot \frac{\partial \tilde{F}}{\partial w_j}\right)^2\right] \right\}^{1/2}
$$

$$
\left. + \left\{ \mathbb{E}\left[\left(\langle \nabla_{Y_j,\cdot}^2 \tilde{F}, \nabla_{Y_j,\cdot}^2 \tilde{F} \rangle\right)^2\right] \cdot \mathbb{E}\left[\left(\frac{\partial \tilde{F}}{\partial Y_j} \cdot \frac{\partial \tilde{F}}{\partial Y_j}\right)^2\right] \right\}^{1/2} \right\}^{1/2}
$$

$$
= c_M \left\{ \sum_{j=1}^{n} 2 \left\{ \mathbb{E}\left[\left(\frac{1}{n}\sigma_w^2 \Gamma^3 w_j \tau'\left(\Gamma Y_j\right) \tau''\left(\Gamma Y_j\right)\right)^2\right] \cdot \mathbb{E}\left[\left(\frac{1}{n}\sigma_w^2 \Gamma w_j \tau\left(\Gamma Y_j\right) \tau'\left(\Gamma Y_j\right)\right)^2\right] \right\}^{1/2} \right.
$$

$$
+ \left\{ \mathbb{E}\left[\left(\frac{1}{\sqrt{n}}\sigma_w \Gamma \tau'\left(\Gamma Y_j\right)\right)^4\right] \cdot \mathbb{E}\left[\left(\frac{1}{\sqrt{n}}\sigma_w \tau\left(\Gamma Y_j\right)\right)^4\right] \right\}^{1/2}
$$

$$
+ \left\{ \mathbb{E}\left[\left(\frac{1}{n}\sigma_w^2 \Gamma^2 \{\tau'\left(\Gamma Y_j\right)\}^2 + \frac{1}{n}\sigma_w^2 \Gamma^4 w_j^2 \{\tau''\left(\Gamma Y_j\right)\}^2\right)^2\right] \right.
$$

$$
\left. \left. \times \mathbb{E}\left[\left(\frac{1}{\sqrt{n}}\sigma_w \Gamma w_j \tau'\left(\Gamma Y_j\right)\right)^4\right] \right\}^{1/2} \right\}^{1/2}
$$

$$
\overset{\mathbb{E}w_j^2=1}{=} \frac{c_M}{n} \sigma_w^2 \left\{ \sum_{j=1}^{n} 2\Gamma^4 \left\{ \mathbb{E}\left[\left(\tau'\left(\Gamma Y_j\right) \tau''\left(\Gamma Y_j\right)\right)^2\right] \cdot \mathbb{E}\left[\left(\tau\left(\Gamma Y_j\right) \tau'\left(\Gamma Y_j\right)\right)^2\right] \right\}^{1/2} \right.
$$

$$
+ \Gamma^2 \left\{ \mathbb{E}\left[\left(\tau'\left(\Gamma Y_j\right)\right)^4\right] \cdot \mathbb{E}\left[\left(\tau\left(\Gamma Y_j\right)\right)^4\right] \right\}^{1/2}
$$

$$
\left. + \left\{ \mathbb{E}\left[\left(\Gamma^2 \{\tau'\left(\Gamma Y_j\right)\}^2 + \Gamma^4 w_j^2 \{\tau''\left(\Gamma Y_j\right)\}^2\right)^2\right] \cdot \mathbb{E}\left[\left(\Gamma w_j \tau'\left(\Gamma Y_j\right)\right)^4\right] \right\}^{1/2} \right\}^{1/2}
$$

$$\overset{iid}{=} \frac{c_M}{\sqrt{n}}\sigma_w^2 \left\{ 2\Gamma^4 \left\{ \mathbb{E}\left[\left(\tau'\left(\Gamma Z\right)\tau''\left(\Gamma Z\right)\right)^2\right] \cdot \mathbb{E}\left[\left(\tau\left(\Gamma Z\right)\tau'\left(\Gamma Z\right)\right)^2\right] \right\}^{1/2}\right.$$

$$+\Gamma^2 \left\{ \mathbb{E}\left[\left(\tau'\left(\Gamma Z\right)\right)^4\right] \cdot \mathbb{E}\left[\left(\tau\left(\Gamma Z\right)\right)^4\right] \right\}^{1/2}$$

$$\left. +\left\{ \mathbb{E}\left[\left(\Gamma^2\left\{\tau'\left(\Gamma Z\right)\right\}^2 + \Gamma^4 w_j^2 \left\{\tau''\left(\Gamma Z\right)\right\}^2\right)^2\right] \cdot \mathbb{E}\left[\left(\Gamma w_j \tau'\left(\Gamma Z\right)\right)^4\right] \right\}^{1/2} \right\}^{1/2}$$

$$= \frac{c_M}{\sqrt{n}}\sigma_w^2 \left\{ 2\Gamma^4 \left\{ \mathbb{E}\left[\left(\tau'\left(\Gamma Z\right)\tau''\left(\Gamma Z\right)\right)^2\right] \cdot \mathbb{E}\left[\left(\tau\left(\Gamma Z\right)\tau'\left(\Gamma Z\right)\right)^2\right] \right\}^{1/2}\right.$$

$$+\Gamma^2 \left\{ \mathbb{E}\left[\left(\tau'\left(\Gamma Z\right)\right)^4\right] \cdot \mathbb{E}\left[\left(\tau\left(\Gamma Z\right)\right)^4\right] \right\}^{1/2}$$

$$+\left\{ \left(\Gamma^4\mathbb{E}\left[\left\{\tau'\left(\Gamma Z\right)\right\}^4\right] + 2\Gamma^6\mathbb{E}\left[\left\{\tau'\left(\Gamma Z\right)\right\}^2\left\{\tau''\left(\Gamma Z\right)\right\}^2\right] + 3\Gamma^8\mathbb{E}\left[\left\{\tau''\left(\Gamma Z\right)\right\}^4\right]\right)\right.$$

$$\left.\left.\times 3\Gamma^4 \cdot \mathbb{E}\left[\left\{\tau'\left(\Gamma Z\right)\right\}^4\right] \right\}^{1/2}\right\}^{1/2}$$

$$= \frac{c_M}{\sqrt{n}}\sigma_w^2 \left\{ 2\Gamma^4 \left\{ \mathbb{E}\left[|\tau'\left(\Gamma Z\right)|^2|\tau''\left(\Gamma Z\right)|^2\right] \cdot \mathbb{E}\left[|\tau\left(\Gamma Z\right)|^2|\tau'\left(\Gamma Z\right)|^2\right] \right\}^{1/2}\right.$$

$$+\Gamma^2 \left\{ \mathbb{E}\left[|\tau'\left(\Gamma Z\right)|^4\right] \cdot \mathbb{E}\left[|\tau\left(\Gamma Z\right)|^4\right] \right\}^{1/2}$$

$$+\left\{ \left(\Gamma^4\mathbb{E}\left[|\tau'\left(\Gamma Z\right)|^4\right] + 2\Gamma^6 \cdot \mathbb{E}\left[|\tau'\left(\Gamma Z\right)|^2|\tau''\left(\Gamma Z\right)|^2\right] + 3\Gamma^8 \cdot \mathbb{E}\left[|\tau''\left(\Gamma Z\right)|^4\right]\right)\right.$$

$$\left.\left.\times 3\Gamma^4 \cdot \mathbb{E}\left[|\tau'\left(\Gamma Z\right)|^4\right] \right\}^{1/2}\right\}$$

where $Z \sim \mathcal{N}(0,1)$. But since $\tau$ is polynomially bounded and the square root is an increasing function, we can bound this expression by

$$\frac{c_M}{\sqrt{n}}\sigma_w^2 \left\{ 2\Gamma^4 \left\{ \mathbb{E}\left[(a + b|\Gamma Z|^\gamma)^4\right] \cdot \mathbb{E}\left[(a + b|\Gamma Z|^\gamma)^4\right] \right\}^{1/2}\right.$$

$$+\Gamma^2 \left\{ \mathbb{E}\left[(a + b|\Gamma Z|^\gamma)^4\right] \cdot \mathbb{E}\left[(a + b|\Gamma Z|^\gamma)^4\right] \right\}^{1/2}$$

$$\left.+\Gamma^4 \left\{ \sqrt{3(1 + 2\Gamma^2 + 3\Gamma^4)} \cdot \mathbb{E}\left[(a + b|\Gamma Z|^\gamma)^4\right] \cdot \mathbb{E}\left[(a + b|\Gamma Z|^\gamma)^4\right] \right\}^{1/2}\right\}^{1/2}$$

$$= \frac{c_M}{\sqrt{n}}\sigma_w^2 \sqrt{\Gamma^2 + \Gamma^4(2 + \sqrt{3(1 + 2\Gamma^2 + 3\Gamma^4)}}\left\{ \mathbb{E}\left[(a + b|\Gamma Z|^\gamma)^4\right] \right\}^{1/2}$$

$$= \frac{c_M}{\sqrt{n}}\sigma_w^2 \sqrt{\Gamma^2 + \Gamma^4(2 + \sqrt{3(1 + 2\Gamma^2 + 3\Gamma^4)}} \cdot \left\| a + b|\Gamma Z|^\gamma \right\|_{L^4}^2,$$

where $Z \sim \mathcal{N}(0,1)$.

