# OpenReview forum: "Non-asymptotic approximations of Gaussian neural networks via second-order Poincar\'e inequalities"
_TMLR — Rejected by TMLR_

### Review · Reviewer_PqrG · 2023-06-12

**Summary Of Contributions:**

Full disclosure: I have reviewed the paper for another venue before. While I do think it is a proper fit for TMLR, some of the comments I gave last time should still be addressed (see below)

The paper proves a nonasymptotic bound for the convergence of finite-dimensional output vectors of two-layer neural networks to Gaussian random vectors. The main tool is the Second order Poincare inequality, introduced by Chaterjee, which quantifies the distance to a Gaussian, in terms of a map's Hessian.

Lately, there has been a lot of interest in the convergence rates of randomly initialized neural networks to Gaussian processes and the paper adds another tool to this growing literature. The main contribution of the paper in this direction is to establish apparently optimal rates of convergence, 1/sqrt{n}, with explicit constants.


**Audience:**

Yes

**Claims And Evidence:**

Yes

**Requested Changes:**


My main requested change is to make the comparisons with the literature more explicit. There are many results of similar flavor and the authors need to emphasize which results, in which settings, are new.

Perhaps the closest result is the recent paper by Basteri and Trevisan. For two layers, as far as I can see Theorem 3.3 in the paper is directly comparable to Theorem 1.1 in Basteri and Trevisan. Both results obtain the rate 1/sqrt(n). However, the current result makes the pre-factor slightly more explicit, with exact dependence on the number of inputs, p. In contrast, the result is only applicable for regular activations and for two layers. If the current paper contains a result that is not covered by Basteri and Trevisan, it should be clearly stated.

Another missed comparison is with a missed paper by Klukowski, who studied convergence rates in functional space and obtained obtain convergence results that do not depend on p at all. In the paper, by Klukowski the convergence rate is also 1/sqrt(n), so it should be comparable as well.

Other than that I have two other comments:
- This is a matter of personal taste, but Theorems 3.1 and 3.2 are special cases of Theorem 3.3. Sometimes it makes sense to present proofs of easy first cases first. But here I did not find the proofs of Theorems 3.1 and 3.2 to be particularly illuminating. Moreover, these results are already known and follow from the classical Berry-Eseen theorem for the Kolmogorv distance and the CLT of Rio for the Wasserstein distance. (Whether the authors decide to keep these proofs or not will not affect my recommendation, but if they do decide to keep them in, it should be stated that the results are not new)

- In Section 5 the authors mention the results can be extended to deep networks. If this is the case it should be stated as a result.

References:

- Rio, E. (2009). Upper bounds for minimal distances in the central limit theorem. In Annales de l'IHP Probabilités et statistiques (Vol. 45, No. 3, pp. 802-817).

- Klukowski, A. (2022). Rate of Convergence of Polynomial Networks to Gaussian Processes. In Conference on Learning Theory (pp. 701-722). PMLR.

**Strengths And Weaknesses:**

Strengths:
- The problem is interesting.
- The approach is natural and useful in this context.

Weakness:
- Some comparisons with the literature are missing. In particular, it is hard to appreciate if the quantitative results are new.
- The results only apply to randomly initialized networks and do not offer insights about trained networks. (This is general criticism against this line of research and I do not hold it against the paper)
- The results only apply to two-layered networks

---

### Review · Reviewer_QP7C · 2023-06-12

**Summary Of Contributions:**

In this paper, the authors focus on single-hidden-layer neural network (NN) model having $n$ neurons with Gaussian weights and biases.
While it is known in the infinitely wide limit (as $n \to \infty$ that the output of such NN converges in distribution to a Gaussian process, it remains unclear the rate of convergence.
In this paper, using second-order Gaussian Poincare inequalities, the authors provide upper bounds on the 1-Wasserstein distance, the total variance distance, as well as the Kolmogorov-Smirnov distance between the network output and the (limiting) Gaussian distribution, in Theorem 3.1 for a 1-dimensional single scalar input, in Theorem 3.2 for a single vector input of dimension d, and in Theorem 3.3 for p input of dimension d.
The obtained bounds are functions of the (properties of the) activation function and the dimensions.
Numerical experiments are provided in Section 4 to validate the theoretical results in Theorem 3.1.

**Audience:**

Yes

**Broader Impact Concerns:**

This paper is mainly theoretical and I do not see any ethnical concerns.

**Claims And Evidence:**

Yes

**Requested Changes:**

Below are some detailed comments.

1. above equation (2) "by the linear envelope": shouldn't it be the polynomial envelope here?
2. in section 1.2: the authors claimed that "provide estimates of the approximation error with optimal rate, and tight constants", is the theoretically or empirically supported?
3. I am not sure whether the detailed derivations of Theorem 3.1 and 3.3 are necessary. I would suggest to replace them by, e.g., discussions on the implications of the obtained results, particularly of Theorem 3.3.
4. As mentioned above, Theorem 3.3 needs more discussions and explanations: how will the data and activation function come into play in the behavior of this distance? Can some special cases be studied in a more explicit manner?
5. Page 10: SLNN for single-layer NN?
6. can Theorem 3.3 be numerically supported in some way?
7. figures in Sec 4: the bounds seems really loose, I am not sure if the trends are more visible in log scale?
8. page 12: non-trivial aft all -> after all?

**Strengths And Weaknesses:**

**Strengths**: applying the technique of second-order Gaussian Poincaré inequalities to characterization of shallow NN is, interesting, and to the best of my knowledge, novel. I did not check the detailed proof (which is in essence burdensome algebraic manipulation) bu the obtains results looking reasonable.

**Weaknesses**: despite derived here using rather advanced techniques, the obtained bounds seem still rather loose in simulations. Also, the model under study is limited to single-hidden-layer NN and the activation is limited to C^2, making the results of limited practical interest (although some limitations are discussed in Sec 5).

---

### Review · Reviewer_r8ym · 2023-06-12

**Summary Of Contributions:**

The authors give explicit bounds on the distance between the prior distribution of a shallow finite-width neural network and its Gaussian equivalent. They utilize a technique based on second-order Poincare inequality, which makes the bounds' constant explicit. The network has biases. The activation function is twice-differentiable (hence ReLU is omitted) and bounded by a polynomial. Numerical bounds seem vacuous for Tanh but track the qualitative behavior for $ x^3 $.

**Audience:**

No

**Broader Impact Concerns:**

---

**Claims And Evidence:**

Yes

**Requested Changes:**

Missing citations:
* Noci, Lorenzo, et al. "Precise characterization of the prior predictive distribution of deep ReLU networks." Advances in Neural Information Processing Systems 34 (2021): 20851-20862.
* Zavatone-Veth, Jacob, and Cengiz Pehlevan. "Exact marginal prior distributions of finite Bayesian neural networks." Advances in Neural Information Processing Systems 34 (2021): 3364-3375.

both papers compute the exact prior in terms of Meijer-G functions. There might be more citations in these papers that are missing.
Given that the exact prior computations are done, including the deep case, I have a hard time appreciating a lot the contributions of this paper. Can the authors please comment on the differences in their contribution to these prior works?

Is it possible to generalize your results to the deep case?


**Strengths And Weaknesses:**

Strengths:
- explicit constants in the bounds
- analysis includes the prior on multiple input points
- the network has biases

Weaknesses:
- it is not clear how to generalize the analysis to deep networks

- although the constant in the bound is explicit, it seems to be vacuous for tanh which is the more interesting case due to universal approximation (with $ x^3 $ the shallow network can only compute a third-order polynomial of the input)

- the full proofs are given in the main paper which takes 4-5 pages. It would be much better to give a sketch (for ex. the computations of mean and covariance) and defer the full proof to the appendix (which would be these computations combined with results from Vidotto 2020 if I'm not missing something).

---

### Decision · Action_Editors · 2023-07-05

**Recommendation:** Reject

**Comment:**

All the reviewers and I agree that the paper presents interesting and novel ideas. However, we believe that further improvements are necessary before it is suitable for publication. Taking into account the reviewers' suggestions and improving the paper's writing and organization, we believe the paper has the potential to be accepted for publication in TMLR. We encourage the authors to thoroughly revise and resubmit the paper.

**Audience:**

The paper focuses on limit theorems for neural networks, making it relevant and interesting for the TMLR audience.

**Claims And Evidence:**

In this paper, the authors focus on establishing non-asymptotic convergence bounds for a sequence of single-hidden-layer neural networks. Specifically, they consider neural networks with increasing width and study their convergence towards a Gaussian limit.

To analyze this convergence, the authors employ second-order Poincaré inequalities.

**Resubmission Of Major Revision:**

The authors may consider submitting a major revision at a later time.